# Is Adnexectomy Mandatory at the Time of Hysterectomy for Uterine Sarcomas? A Systematic Review and Meta-Analysis

**DOI:** 10.3390/medicina58091140

**Published:** 2022-08-23

**Authors:** Carlo Ronsini, Aniello Foresta, Matteo Giudice, Antonella Reino, Marco La Verde, Luigi della Corte, Giuseppe Bifulco, Pasquale de Franciscis, Stefano Cianci, Vito Andrea Capozzi

**Affiliations:** 1Department of Woman, Child and General and Specialized Surgery, University of Campania “Luigi Vanvitelli”, 80138 Naples, Italy; giudice.matteo@yahoo.it (M.G.); antonella.reino@studenti.unicampania.it (A.R.); marco.laverde88@gmail.com (M.L.V.); pasquale.defranciscis@unicampania.it (P.d.F.); 2Department of Woman and Child Sciences, Fondazione Policlinico Universitario A. Gemelli, IRCCS, Università Cattolica del Sacro Cuore, 00168 Roma, Italy; anielloforesta@gmail.com; 3Department of Neuroscience, Reproductive Sciences, and Dentistry, University of Naples Federico II, 80138 Naples, Italy; giuseppe.bifulco@unina.it; 4Department of Obstetrics and Gynecology, University of Messina, 98122 Messina, Italy; stefanoc85@gmail.com; 5Department of Medicine and Surgery, University Hospital of Parma, 43125 Parma, Italy; capozzivitoandrea@gmail.com

**Keywords:** bilateral salpingo-oophorectomy, uterine stromal sarcoma, uterine adenosarcoma, uterine leiomyosarcoma, endometrial stromal sarcoma

## Abstract

***Background and Objectives:*** Uterine sarcomas represents only 3% of all the female genital tract ones. The tumoral stage is the most significant prognostic factor. The role of the bilateral salpingo-oophorectomy (BSO) in the surgical management of FIGO stage IA and IB appears still controversial. This review aims to investigate the impact of bilateral adnexectomy in the treatment of uterine sarcoma. ***Methods:*** Following the recommendations in the Preferred Reporting Items for Systematic Reviews and Meta-Analyses (PRISMA) statement, we systematically searched the PubMed, Scopus, Cochrane, Medline, and Medscape databases in February 2022. We applied no language or geographical restrictions, but we considered only English studies. We included the studies containing data about Recurrence Rate (RR), Disease-free Survival (DFS), and Overall Survival (OS). We used comparative studies for meta-analysis. ***Results:*** Seventeen studies fulfilled the inclusion criteria; 2 retrospective observational studies, and 15 retrospective comparative studies, And 14 out of the 15 comparative studies were enrolled in meta-analysis. A total of 3743 patients were analyzed concerning the use of adnexectomy with hysterectomy in patients with uterine sarcoma and compared with those who did not. Meta-analysis highlighted a non-significant worsening of the OS in the BSO group compared to the OP group and showed that adnexectomy does not improve the DFS (BSO OR 1.23 (95% CI 0.81–1.85) *p* = 0.34; I^2^ = 24% *p* = 0.22). ***Conclusions:*** Most studies selected for our review showed that adnexectomy does not significantly affect the RR, OS, and PFS in treating FIGO stage I uterine sarcomas. Therefore, even if there is a unanimous consensus about bilateral adnexectomy in menopausal patients, preservation of ovarian tissue may be considered in premenopausal women. Nonetheless, there are not enough cases in the literature to recommend this procedure.

## 1. Introduction

Uterine sarcomas are rare cancers representing 3–7% of all uterine cancers and only 3% of all female genital tract ones. The mean age at diagnosis is 56 years. They are classified into 4 histological types: leiomyosarcomas (60%), endometrial stromal sarcomas (10–15%), undifferentiated sarcomas (5–10%) and adenosarcomas (10%) [1]. The FIGO staging of uterine sarcomas identifies 4 different stages.

The subject of our study is uterine sarcomas at FIGO stage IA (tumor limited to the uterine body with a size <5 cm for leiomyosarcomas and limited to the endometrium/endocervix without myometrial invasion for endometrial stromal sarcoma/adenosarcoma) or IB (tumor limited to uterus >5 cm for leiomyosarcomas and invading less than half a myometrium for endometrial stromal sarcomas/adenosarcoma). 

According to the NCCN guidelines, the adnexectomy can be “considered” during a total abdominal hysterectomy (TAH) in FIGO IA and IB sarcomas but does not represent a standard treatment. 

Beyond the stage, the other prognostic factors that affect the therapeutic choice are the histotype, the grade, and the patient’s age. However, in the surgical approach, the role of the bilateral salpingo-oophorectomy (BSO) in treating uterine sarcomas confined to the uterus (FIGO stage IA and IB) is still controversial. This review investigates the role of bilateral adnexectomy in the surgical treatment of uterine sarcomas.

## 2. Materials and Methods

The methods for this study were specified a priori based on the recommendations in the Preferred Reporting Items for Systematic Reviews and Meta-Analyses (PRISMA) statement [2]. We required registration to the PROSPERO site for meta-analysis with protocol number 315393.

### 2.1. Search Method

We conducted a systematic search for studies about the addiction of adnexectomy to hysterectomy in patients with uterine sarcomas in the PubMed, Cochrane, Medline, and Medscape databases up to February 2022. We included the studies from the earliest publication reported in the scientific literature with no restriction on the country. We considered only English entirely published studies. 

### 2.2. Study Selection

Study selection was made independently by AR and AF. In case of discrepancy, a third person (CR) was questioned about its inclusion or exclusion. Inclusion criteria were: (1) studies that included patients with FIGO stage IA or IB uterine sarcomas; (2) studies that reported at least one outcome of interest (RR, OS, PFS); (3) peer-reviewed articles published initially been. We excluded non-original studies, preclinical trials, animal trials, abstract-only publications, and articles in a language other than English. When possible, the authors of studies, only published as congress abstracts, were contacted via email and asked to provide their data. The studies selected and all reasons for exclusion are mentioned in the Preferred Reporting Items for Systematic Reviews and Meta-Analyses (PRISMA) flowchart (Figure 1). All included studies were assessed regarding potential conflicts of interest. 

### 2.3. Statistical Analysis 

The risk rate (RR) and 95% confidence intervals (CI) were used for dichotomous variables. RR, DFS, and OS were used as clinical outcomes. In each study, the Recurrence Rate was defined as the percentage of relapse after the primary treatment. Disease-free survival was defined as the time elapsed between surgery and recurrence or the date of the last follow-up. Overall survival has been defined as the time elapsed between surgery and death of disease or the last follow-up. Review Manager version 5.4.1 (REVman 5.4.1) and IBM Statistical Package for Social Science (IBM SPSS version 25.0) for MAC were used for statistic calculation. Fixed-effect models conducted statistical analysis without significant heterogeneity (I^2^ < 50%) or random-effect models if I^2^ > 50%. For all performed analyses, a *p*-value < 0.05 was considered significant. 

### 2.4. Quality Assessment

The quality of the included studies was assessed using the Newcastle–Ottawa scale (NOS) [3]. This assessment scale uses three broad factors (selection, comparability, and exposure), with the scores ranging from 0 (lowest quality) to 8 (best quality). Two authors (A.R. and A.F.) independently rated the study’s quality. Any disagreement was subsequently resolved by discussion or consultation with NC. The NOS scale is reported in Appendix A Table A1.

## 3. Results

### 3.1. Studies’ Characteristics

After the database search, 494 articles matched the searching criteria. After removing records with no full text, duplicates, and wrong study designs (e.g., reviews), 38 were suitable for eligibility. Of those, 17 matched the inclusion criteria and were included in the systematic review. Two of them were non-comparative, single-armed studies evaluating only TAH. Fifteen were comparative studies between BSO and Ovarian Preservation (OP). Fourteen of them were included in the quantitative analysis (Figure 1). We summarized the countries where the studies were conducted, the publication year range, the studies’ design, the number of participants, and the median follow-up in Table 1. The quality of all studies was assessed by NOS [3]. Overall, the publication years ranged from 1990 to 2021. In total, 3743 patients diagnosed with FIGO stage I uterine sarcoma were enrolled. The mean follow-up period ranged from 32 to 123 months.

### 3.2. Outcomes

A total of 3743 patients were included in the review. Twelve out of the 17 selected studies presented RR data. Except for 4, the other 13 studies did not present OS data. By alphabetic analysis, 15 studies performed a retrospective comparison between BSO and OP in patients diagnosed with FIGO stage I uterine sarcoma. In summary, no study highlighted a statistically significant difference in RR, OS, and DFS between patients who underwent BSO and those who did not. We summarized these results in Table 1.

#### 3.2.1. Meta-Analysis

Fourteen out of the 15 comparative studies comparing BSO and OP were enrolled in the meta-analysis. A total of 3679 patients were analyzed: 1388 patients in the OP arm were compared with 2291 patients who underwent BSO, exploring RR, OS, and DFS as outcomes. The fixed-effects model was applied because of the low heterogeneity (I^2^ < 50%; *p* = 0.22).

BSO group showed a non-significant worsening of the OS compared to the OP group (OR 1.20 (95% CI 0.99–1.46) *p* = 0.06) (Figure 2; Appendix B Figure A1).

As well, in our analysis, the ovarian preservation led to a better DFS (BSO OR 1.23 (95% CI 0.81–1.85) *p* = 0.34; I^2^ = 24% *p* = 0.22). (Figure 3; Appendix B Figure A2).

About this outcome, we performed a sub-analysis by histotype of uterine sarcoma. Only 12 of the 14 comparative studies enrolled reported valuable data, with 332 patients for the ESS group, 114 for the LMS group, and 24 for the AS group.

#### 3.2.2. Endometrial Stromal Sarcoma (ESS)

Several studies have investigated the effect of adnexectomy in the treatment of endometrial stromal sarcoma on RR, OS, and PFS. 

In the Italian study published by Gadducci et al. [8] in 1995, the clinical histories of 66 patients with endometrial stromal sarcoma treated from 1980 to 1994 at the Department of Gynecology and Obstetrics of the University of Brescia, Monza, Padua, Pisa, and Turin were analyzed. In this study, the sample was divided into 2 groups (low-grade and high-grade tumors), subsequently stratified by stage and age. It was found that, among the 6 patients who underwent a total abdominal hysterectomy (TAH) with a bilateral salpingo-oophorectomy (BSO) and the other 6 ones with residual ovarian tissue, all of them under the age of 50 and with a diagnosis of a FIGO stage I sarcoma, the recurrence rate was, respectively, 33.3% and 16.7% (*p* = NS). 

A retrospective study by A. Berchuck et al. [4] was led, from 1970 to 1984, to treat endometrial stromal tumors. Two of them had a relapse among the 6 patients affected by stage I endometrial stromal sarcoma who underwent a bilateral ovariectomy. 

In the clinical study by F. Amant et al. [6], among 18 women of premenopausal age diagnosed with stage I ESS, 12 underwent TAH+BSO and 6 TAH or a radical abdominal hysterectomy (RAH) without BSO. The median age at diagnosis was 44 years (18–60 years). The relapse occurred in 3 out of 12 (25%) and 1 out of 6 (17%), respectively (differences = 8.3%, 95% CI = 34.5 to 39.7%). 

In the retrospective study by Ning Li et al. [14], 53 patients with a median age of 44 at diagnosis were selected for the study. Thirty-seven patients had a low-grade ESS, 11 an undifferentiated endometrial sarcoma (UES), and 5 had unclassified ESS. All patients received as initial therapy TAH/RAH, and in 9 patients, ovary function was preserved.

The recurrence rates of the patients with and without preserving ovarian function at primary treatment were 100% and 22.7%, respectively (*p* < 0.001). The 2-year and 5-year survival rates were 91.5% and 85.9%, respectively. Three of 5 deceased cases underwent TAH+BSO, and 2 underwent TAH only.

The retrospective study by Kim WY et al. [11] was led on 22 patients with a median age of 43 years. The median follow-up was 77 months. They all underwent a hysterectomy, and BSO was performed in 11 cases. Disease relapsed in 5 out of 11 (45.4%) patients treated with BSO. The disease-free survival was not different in the 2 groups of patients, those who underwent BSO and those who did not (110 and 120 respectively; *p* = 0.5045).

In the study by Volkan Karataşlı et al. [10], 7 women (21%) with stage I low-grade ESS underwent a hysterectomy, while in 27 of them (79%), a hysterectomy with BSO was performed. Disease relapsed in 1 patient in the first group and 3 patients in the second one. All recurred cases were at stage IB. No significant difference in RR was detected in this cohort between the two groups (odds ratio (OR) 1.14, 95% confidence interval (CI) 0.12–11.10, *p* = 0.912). No significant differences in OS or PFS were observed between the ovarian preservation and BSO groups (*p* = 0.235 and *p* = 0.688, respectively). 

In the study by Nasioudis, D. et al. [15], 743 patients who met the inclusion criteria were identified; 541 (72.8%) had BSO. There was no difference in OS between patients who did and did not undergo BSO, *p* = 0.50; 5-year OS rates were 96.2% and 97.1%, respectively.

The performance of BSO was not associated with better survival (HR: 1.28, 95% CI: 0.51, 3.19).

Stewart, L.E. et al. [19] identified 47 cases of stage I LG-ESS. Thirty-seven (52%) patients underwent BSO. Among those who underwent BSO at time of diagnosis, 35% experienced a recurrence compared to 66% of recurrence among women who preserved their ovaries (*p* = 0.051). PFS was 38 vs. 11 months for those who underwent BSO vs. those who retained their ovaries (*p* = 0.55).

The practice of BSO showed a prolongation of both PFS (38 vs. 11 months) and OS (45 vs. 14 months) in those who underwent an oophorectomy at diagnosis. The risk of recurrence in the BSO group was 35% compared to 66% in the no BSO group (*p* = 0.051).

A subgroup analysis was carried out according to menopausal status to determine the impact of ovarian preservation on the RFS. In premenopausal patients, the 5-year RFS rate of patients who underwent BSO was significantly higher than those who did not (95.5 vs. 59.6%; *p* = 0.018). On the other side, in the menopausal patients, although it did not reach statistical significance, women who underwent BSO showed a higher 5-year RFS rate than those who did not (80.0% vs. 57.1%; *p* = 0.063).

Li, A.J. et al. [13] focused their study on a cohort of patients at stage I low-grade ESS treated in 5 different American institutes from 1976 to 2002. Twelve premenopausal patients who did not undergo BSO were matched to 24 controls. They found a lower RR in the case group but it was not statistically significant (RR: 33% vs. 42%, *p* = 0.63).

In the study by Nasioudis, D. et al. [16], 520 out of 1482 patients (35.1%) were the cases of low-grade endometrial stromal sarcoma (LG-ESS).

OP was documented in 418 women (28.2%). The OP rate in patients with a diagnosis of LG-ESS was 29.0%.

In the whole cohort of patients considered, there was no difference in overall (*p* = 0.220) or cancer-specific (*p* = 0.210) survival (CSS) between women who underwent an oophorectomy and those who did not. No difference in survival was noted for women with stage IA disease (OS: *p* = 0.220; CSS: *p* = 0.470). However, a statistically significant survival was found for those at stage IB who had an OP (OS: *p* = 0.044; CSS: *p* = 0.040). 

Specifically, in patients with LG-ESS, the preserved ovaries had comparable OS (*p* = 0.410) and CSS (*p* = 0.560) rates but were not statistically significant.

Those results are summarized in Table 2.

In our meta-analysis, OP showed a favorable impact on DFS in this subgroup of patients (BSO OR 2.04 (95% CI 1.01–4.12) *p* = 0.05; I^2^ = 27% *p* = 0.22) (Figure 3; Appendix B Figure A2).

#### 3.2.3. Leiomyosarcoma (LMS)

In the study by H.K. Sait et al. [15], patients with histologically proven uterine sarcoma at stage I (7 carcinosarcomas, 5 leiomyosarcomas, and 3 undifferentiated endometrial sarcomas) were enrolled. The surgical procedure adopted for them was hysterectomy (H)+BSO, with a story of relapse in only 2 patients.

L.T. Soh et al. [18] focused their study on patients with stage I high-grade leiomyosarcoma diagnosed and treated between 1997 and 1998. Three women had a total hysterectomy (TH), while 13 women had TH+BSO. All patients treated with only TH relapsed. The disease relapsed in 6 of 13 women who underwent TH+BSO.

A total of 793 patients were identified by Sia, T.Y. et al. [17]. Overall, 225 patients (28.4%) had OP. The 5-year survival for the OP group was 67.1% (95% CI 59.8–75.2%), compared to 72.2% for the BSO group (95% CI 67.2–77.5%). In stage IA patients, 5-year survival was 89.4% (95% CI 77.6–100.0%) in the OP group compared to 76.5% for women who underwent an oophorectomy (95% CI 63.9–91.6%). In stage IB tumors, the 5-year survival was 58.1% in the OP group (95% CI 48.8–69.0%), compared to 70.1% in the BSO group (95% CI 64.1–76.6%).

In the study by Nasioudis, D. et al. [16], among 1482 patients, there were 800 (54.0%) cases of LMS. In the LMS group, although women with OP had better 5-year OS (72.8% vs. 68.9%) and CSS (74.2% vs. 70.8%) rates, this difference did not reach statistical significance (OS: *p* = 0.078; CSS: *p* = 0.098). 

Gadducci, A. et al. [7], among stage I patients younger than 50 years who underwent a total abdominal hysterectomy, found that relapse occurred in 33.3% of the 21 patients who had bilateral salpingo-oophorectomy and in 23.8% of the 21 patients who were left with one or both ovaries (*p* = NS)

In the study by Kapp et al. [9], 240 patients aged <50 years at stage I or II underwent BSO. These patients had similar survival compared to that of 101 women (aged <50 years, stage I and II) who did not undergo an oophorectomy (5-year disease-specific survival-DSS rate of 83.2% vs. 83.2%; *p* = 0.445).

There was no statistically significant difference in the 5-year DSS for patients who underwent an oophorectomy at their initial cancer surgery compared to those who did not (66.2% vs. 72.3%, respectively; *p* = 0.150). 

Giuntoli et al. [19] identified 2 groups of patients diagnosed with LMS treated at the Mayo Clinic from 1976 to 1999. The first group of 25 patients (23 at stage I, 1 at stage II, and 1 at stage IV) who maintained ovarian function represented the case group; the second one of 25 patients (22 at stage I, 2 at stage II, and 1 at stage IV) with documented BSO was the control group. No difference was detected in the recurrence rate.

OP did not seem to affect the outcome in a univariate analysis adversely. Patients who did not undergo BSO had significantly better disease-specific survival than patients who had their ovaries removed.

Because of the unexpected association between OP and improved survival, this study explored this relationship further with a case-control study. The case group was comprised of 25 patients who maintained ovarian function. The 25 cases were matched by stage, grade, and age to 25 controls with documented BSO. 

The 25 cases with ovarian preservation showed no significant difference in disease-specific survival compared with the 25 controls who underwent an oophorectomy. Additionally, cases and controls demonstrated no difference in the risk of recurrence.

Those results are summarized in Table 2.

Our meta-analysis did not show any significant difference in DFS between patients who did or did not undergo BSO (adnexectomy OR 2.04 (95% CI 1.01–4.12) *p* = 0.05; I^2^ = 27% *p* = 0.22) (Figure 3; Appendix B Figure A2).

#### 3.2.4. Adenosarcoma (AS)

In the retrospective study by Young Jae-Lee et al. [12], among 31 patients 44 years with a diagnosis of Mullerian adenosarcoma, 24 underwent a hysterectomy; only 13 performed adnexectomy. The study has not shown a result that is statistically significant in terms of risk of recurrence.

In the multicentric retrospective study by Nasioudis, D. et al. [16], the rate of OP in women with AS was low (18.5%) compared to patients with a diagnosis of LMS or LG-ESS. Therefore, there was not a statistically significant difference in OS (*p* = 0.350) and in Cancer-specific Survival (CSS) (*p* = 0.560) among the group who underwent BSO and the one who did not.

Those results are summarized in Table 2.

Only one comparative study was eligible for histotype stratification of our meta-analysis, with no statistically significant role of BSO on DFS (OR 0.48 (95% CI 0.04–1.63) *p* = 0.56) (Figure 3; Appendix B Figure A2).

## 4. Discussion

The surgery represents the primary treatment for early-stage uterine sarcomas: it includes a total abdominal hysterectomy with or without bilateral salpingo-oophorectomy [21,22]. Uterine sarcomas often express hormonal receptors, so removing the adnexa can hypothetically reduce the hormone-dependent proliferative stimulus on the neoplasm itself. Davidson et al. [23] showed that estrogen (ER) and progesterone receptors (PR) are expressed in 47% and 63% of all the uterine sarcomas, respectively. Furthermore, removing the adnexa would allow reducing metastatic foci. Pelvic relapse in early-stage uterine sarcomas occurs in 14–34% of the patients after 5 years [24], while ovarian metastases only in 3.9% [25]. Our metanalysis highlighted that the RR ranged from 7.7% to 100% in patients who underwent BSO; vice versa, in those with OP, between 17% and 66%. However, in premenopausal patients who underwent surgical castration, menopause symptoms and the potential increased risk for coronary heart diseases, hip fractures, and strokes can impact their quality of life. Besides, it should be considered that the height of hormonal receptor expression, particularly in low-grade ESS, is related to a less aggressive clinical behavior and a more favorable prognosis, as shown by Leitao et al. [25] and Chan et al. [26]. By qualitative and quantitative analysis of the data, our study showed that removing the adnexa does not improve the OS (Figure 3) and that the ovarian preservation causes a slight increase in the DFS, especially in women with a diagnosis of ESS (Figure 3). However, the tumor’s staging, grading, and histological type must be considered in the therapeutic choice. Reassuring data come from the studies on low-grade endometrial stromal sarcoma patients who underwent an adnexectomy.

In deciding on performing an oophorectomy in younger patients, the potential increased risks for coronary heart disease, hip fracture, and stroke associated with oophorectomy in patients aged <65 years must be considered. Parker et al. [21] used a Markov decision analysis model to estimate the optimal strategy for maximizing survival for women at average risk of ovarian cancer.

Their model predicted that a BSO performed in women aged <55 years would result in an 8.58% excess mortality by age 80 years, whereas women who underwent BSO aged <59 years would have 3.92% excess mortality. The predicted excess in mortality from a BSO persisted until age 75. Their analysis suggests that the ovaries should be preserved in women until at least age 65.

As for the adnexectomy, the role of adjuvant chemotherapy and local radiotherapy appears still controversial too. Furthermore, our analysis showed that the choice of surgical and adjuvant treatment could not depend only on the stage of the tumor but also on the histological subtype. These considerations attempt to weigh the role of adnexectomy in treating uterine sarcoma as being extremely tricky. Nowadays, the removal of patients with a diagnosis of endometrial stromal sarcoma, including low-grade endometrial stromal sarcoma (LG-ESS) and high-grade endometrial stromal sarcoma (HG-ESS) of the adnexa, should be recommended in those with the high-grade form. In contrast, this option is still questionable in the low-grade ones [4]. It is also necessary to underline how the differentiation between HGSS and undifferentiated uterine sarcoma is only genetic. A histological diagnosis in such cases is not enough. In the past, the majority of the diagnosed undifferentiated uterine sarcomas were actually HGSS. Ovarian preservation does not seem to impact DFS and RR in women younger than 50 with stage I leiomyosarcoma [18]. In these two previous studies, adjuvant chemotherapy for the treatment of stage I sarcomas does not seem to impact clinical outcomes. By the way, as shown in our research, existing data about BSO may be confounding. The study of Kapp et al. [9] revealed no statistically significant difference in the 5-year DFS for patients who did or did not undergo an oophorectomy (66.2% vs. 72.3%; *p* = 0.150). For Nasioudis et al. [15], BSO was not associated with better survival (HR: 1.28, 95% CI: 0.51, 3.19). However, other studies by Amant et al. [6], Kim et al. [11], Karataşlı et al. [10], Li et al. [13], and Giuntoli et al. [19] observed similar recurrence rates in patients who did or did not undergo BSO and the identical DFS among the two groups. Otherwise, in the study by Ning Li et al. [14], the recurrence rate of patients who underwent BSO was significantly lower compared to patients with ovarian function preserved (*p* < 0.001); among the patients who relapsed, 10 cases developed a local recurrence, 7 cases distant metastasis and 2 patients both local and distant recurrence. Stewart et al. [19] showed similar recurrence rates in patients who did or did not undergo BSO but demonstrated a prolongation of both DFS (38 vs. 11 months) and OS (45 vs. 14 months) in those women who underwent BSO; no difference in DFS in those who received adjuvant hormonal therapy. A multivariate analysis by Giuntoli et al. [19] highlighted the potential influence of several risk factors on disease-specific survival. The most important factors are high grade, advanced stage, and BSO; among these, the grade appeared to be the most influential factor (Relative Risk = 6.05). The high grade and a deep myometrial invasion are other important risk factors for recurrence, no matter the surgical strategy [14]. Moreover, older age was the only independent poor prognostic factor for progression-free survival [13]. Other factors associated with disease recurrence in patients with an accidental diagnosis of early-stage LG-ESS are the performance of myomectomy, positive margins, and OP [19]. Ning Li et al. [14] reported how the high recurrence rate correlates to deep myometrial invasion, and the local control rates were better in patients who received pelvic irradiation, especially in patients with LG-ESS and deep myometrial invasion. Recent studies have found a link between TP53 and ATRX mutations with a poor prognosis in LMS and a correlation between JAZF1 rearrangement and metastasis in LG-ESS [27]. 

### Strengths and Limitations

As demonstrated by our analysis, the samples analyzed in the included studies appear limited and highly heterogeneous in most cases. Therefore, the best therapeutic strategy that can drastically reduce the recurrence rate and improve the clinical outcome of these rare and aggressive tumors has not been defined yet. However, it is possible to state the non-superiority of BSO in the treatment of stage I uterine sarcoma, regardless of the histological type, particularly in ESS patients. In any case, one of the strengths of our study is the temporal breadth of the analysis and the systematic nature of the research, making it, to our knowledge, the most comprehensive study on the subject. Despite this, however, from the data in the literature, it was not possible to stratify the sample according to characteristics that might represent bias. For example, it would have been helpful to have a stratification of premenopausal patients and more information on adjuvant therapies used. These data are undoubtedly due to the predominance of retrospective studies. However, the non-inferiority of OP versus BSO may lay the foundation for future randomized clinical trials.

## 5. Conclusions

Most of the studies selected for our review showed that in the treatment of FIGO stage I uterine sarcomas, adnexectomy does not significantly affect the RR, OS, and PFS. 

Therefore, even if there is a pretty unanimous consensus about BSO in menopausal patients, preservation of ovarian tissue may be considered, especially in premenopausal women.

Nonetheless, there are not enough cases in the literature to recommend this procedure.

## Figures and Tables

**Figure 1 medicina-58-01140-f001:**
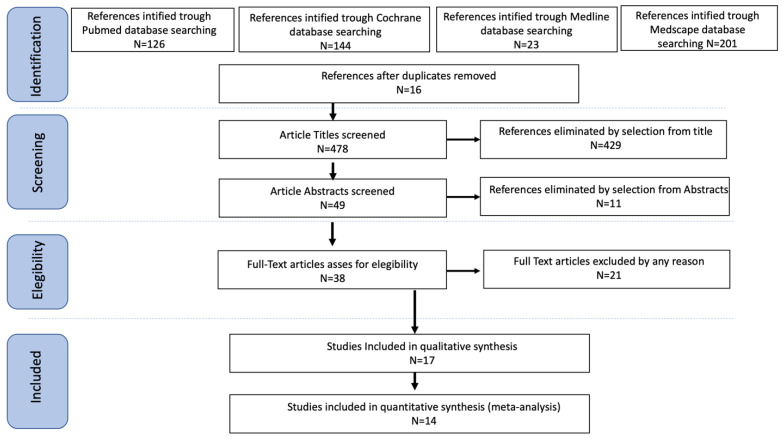
Prisma Flow Chart.

**Figure 2 medicina-58-01140-f002:**
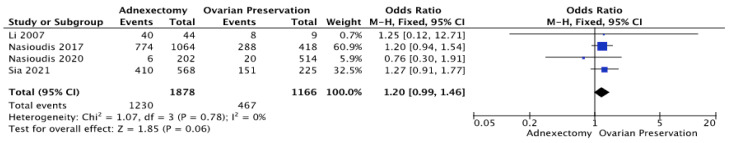
Overall Survival Forrest Plot [14,15,16,17].

**Figure 3 medicina-58-01140-f003:**
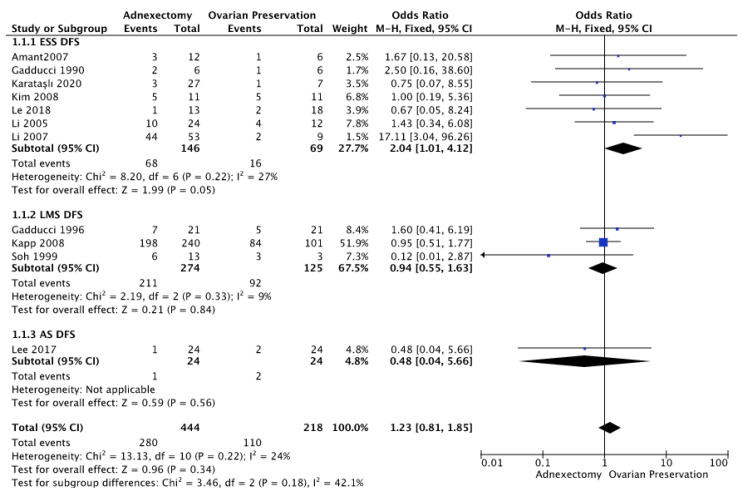
Disease-Free Survival Forrest Plot [6,7,8,9,10,11,12,13,14,18].

**Table 1 medicina-58-01140-t001:** Study characteristics.

Name	Country	Study Design	FIGO Stage I/Population	No. of Participants	Mean FUP Months
Berchuck 1990 [4]	USA	Observational Monocentric Retrospective Cohort study	6/6 ESS (H + BSO ^a^). 2/6 (33%) relapse	12	81
Sait 2014 [5]	Saudi Arabia	Observational Monocentric Retrospective Cohort study	2/15 (13.3%) uterine sarcoma relapse (H + BSO ^a^)	15	24
*Comparative Studies*					
**Name**	**Country**	**Study Design**	**FIGO Stage I/Population**	**No. of Participants**	**Mean FUP Months**
Amant 2007 [6]	Belgium	Observational Multicenter Retrospective Cohort study	18/34 ESS (12 H + BSO ^a^, 6 OP ^b^). RR ^c^: 25% vs. 17% respectively	18	62 and 64 (women with or without recurrence, respectively)
Gadducci 1990 [7]	Italy	Observational Multicenter Retrospective Cohort study	2/6 (33%) LG-ESS ^d^ relapse (H + BSO ^a^) 1/6 (16.67%) LG-ESS ^d^ relapse (OP ^b^)	12	92
Gadducci 1996 [8]	Italy	Observational Multicentric Retrospective Cohort study	88/126 leiomyosarcoma RR ^c^ 33.3% (H + BSO ^a^) 23.8% (OP ^b^)	88	80
Kapp 2008 [9]	USA	Observational Multicentric Retrospective Cohort study	240/341 leiomyosarcoma (H + BSO ^a^) 5 yr DSS ^e^ 83.2% 101/341 leiomyosarcoma (OP ^b^) 5 yr DSS ^e^ 83.2%	341	Not known
Karataşlı 2020 [10]	Turkey	Observational Monocentric Retrospective Cohort study	1/7 LG-ESS ^d^ relapse (H + BSO ^a^) 3/27 LG-ESS ^d^ relapse (OP ^b^)	34	109
Kim 2008 [11]	Korea	Observational Monocentric Retrospective Cohort study	5/11 (45.4%) LG-ESS ^d^ relapse (H + BSO ^a^); DFS ^f^ 110mo 5/11(45.4%) LG-ESS ^d^ relapse (OP ^b^); DFS ^f^ 120mo	22	77
Lee 2017 [12]	Korea	Observational Monocentric Retrospective Cohort study	1/13 uterine adenosarcoma relapse (H + BSO ^a^) 2/18 uterine adenosarcoma relapse (OP ^b^)	31	32
Li 2005 [13]	USA	Observational Multicentric Retrospective Cohort study	10/24 (42%) LG-ESS ^d^ (H + BSO ^a^) PFS ^g^ 91.3 mo 4/12 (33%) relapse (OP ^b^) PFS ^g^ 68.6 mo	36	33 (case) 60 (control)
Li 2007 [14]	China	Observational Monocentric Retrospective Cohort study	10/44 ESS ^d^ relapse (H + BSO ^a^) (RR ^c^ 22.7%) 3/44 deaths (H + BSO ^a^) 9/9 LG-ESS ^d^ relapse (OP ^b^) (RR ^c^ 100%) 2/9 deaths (OP ^b^)	53	66
Nasioudis 2017 [15]	USA	Observational Multicentric Retrospective Cohort study	LMS ^h^: 238 (OP ^b^), 562 (H + BSO ^a^) 5 yr OS ^l^ (72.8% vs. 68.9%) LG-ESS ^d^: 151 (OP ^b^), 369 (H + BSO ^a^) comparable OS ^l^ (*p* = 0.410) AS ^i^: 30 (OP ^b^), 132 (H + BSO ^a^) no difference in OS ^l^ (*p* = 0.350)	1482 patients; 800 (54.0%) cases of LMS; 520 (35.1%) cases of LG-ESS; 162 (10.9%) cases of AS	94 (OP) 123 (BSO)
Nasioudis 2020 [16]	USA	Observational Multicentric Retrospective Cohort study	LG-ESS ^d^: 6/202 deaths (OP ^b^), 5 yr OS ^l^ 97.1% 20/541 deaths (H + BSO ^a^) 5 yr OS 96.2%	743	79.97 (OP) 64.99 (BSO
Sia 2021 [17]	USA	Observational Multicentric Retrospective Cohort study	151/225 alive (OP ^b^), 5 yr OS ^l^ 67.1% 410/568 alive (H + BSO ^a^) 5 yr OS ^l^ 72.2%	793	Not known
Soh 1999 [18]	Singapore	Observational Monocentric Retrospective Cohort study	6/13 (46%) leiomyosarcoma relapse (H + BSO ^a^) 3/3 (100%) leiomyosarcoma relapse (OP ^b^)	16	Not known

(^a^) H+BSO hysterectomy + bilateral salpingo-oophorectomy; (^b^) OP ovarian preservation; (^c^) RR relapse rate; (^d^) ESS, endometrial stromal sarcoma (LG, low grade); (^e^) DSS disease-specific survival; (^f^) DFS disease-free survival; (^g^) PFS progression-free survival; (^h^) LMS, leiomyosarcoma; (^i^) AS, adenosarcoma; (^l^) OS, overall survival.

**Table 2 medicina-58-01140-t002:** Recurrence rate, Overall survival and Disease-Free Survival of the studies.

Study	Population	BSO RR	OP RR	BSO OS	OP OS	BSO DFS	OP DFS
Amant 2007 [6]	18 patients with ESS	25%	17%	/	/	/	/
Berchuck 1990 [4]	6 patients with ESS	33.3%	/	/	/	/	/
Gadducci 1990 [7]	12 patients with ESS	33.3%	16.7%	/	/	/	/
Gadducci 1996 [8]	42 patients with LMS	33.3%	23.8%	/	/	/	/
Giuntoli 1998 [20]	180 patients with LMS	/	/	/	/	88%	92%
Kapp 2008 [9]	341 patients with LMS	/	/	/	/	83.2%	83.2%
Kim 2008 [11]	22 patients with ESS	45.4%	/	/	/	/	/
Karataşlı 2020 [10]	34 patients with ESS	11.1%	14.2%	/	/	/	/
Lee 2017 [12]	24 patients with AS	7.7%	18.1%	/	/	/	/
Li 2005 [13]	36 patients with ESS	42%	33%	/	/	/	/
Li 2007 [14]	62 patients with ESS	100%	22.7%	91.5%	85.9%	/	/
Nasioudis 2017 [15]	1482 patients with ESS, LMS, AS	/	/	68.9%	72.8%	/	/
Nasioudis 2020 [16]	681 patients with ESS	/	/	96.2%	97.1%	/	/
Sait 2014 [5]	15 patients with CS, LMS, US	13.3%	/	/	/	/	/
Sia 2021 [17]	793 patients with LMS	/	/	72.2%	67.1%	/	/
Soh 1999 [18]	16 patients with LMS	46.1%	/	/	/	/	/
Stewart 2018 [19]	47 patients with ESS	35%	66%	/	/	/	/

## Data Availability

Not applicable.

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
