# Peer review of "Is Adnexectomy Mandatory at the Time of Hysterectomy for Uterine Sarcomas? A Systematic Review and Meta-Analysis"

_medicina, 2022, doi:10.3390/medicina58091140_

Round 1

Reviewer 1 Report

Well researched and previous literature extensively reviewed by the authors.

Author Response

Dear Reviewer,

Thank You for taking the time to review our manuscript and for your comments. They are crucial and valuable to us in raising the quality standard of our work.

We wanted to inform You that we have made a general revision of the English and grammar. 

Also, you can find the rewritten and corrected version of the manuscript in the attached file. We highlighted any changes made.

Thank you very much for your advice and comments. We hope we have complied with your requests.

Reviewer 2 Report

The article is interesting as it tries to deal with a debatable question. The authors correctly used the PRISMA statement, which is mandatory for systematic reviews. 

Minor remarks 

It should be stated that the differentiation between HGSS and undifferentiated uterine sarcoma is only genetic. A histological diagnosis in such cases is not enough. In the past, the majority of the diagnosed undifferentiated uterine sarcomas were actually HGSS. Therefore, the articles which investigated undifferentiated uterine sarcomas and you cited ( references -7, 15) are old and this could not be undifferentiated uterine sarcomas, but HGSS. It is better to mention in the abstract, that undifferentiated uterine sarcomas are too rare and aggressive tumors, and there is not enough data to make such conclusions. Therefore, remove uterine sarcomas in the title and abstract and instead use - ULMS, LGSS, and HGSS (without undifferentiated uterine sarcomas). 

Use the final sentence of the conclusion at the end of the abstract - "Nonetheless, there are not enough cases in the literature to recommend this procedure".

- Please change the references list according to the MDPI guidelines

Author Response

Dear Reviewer,

Thank You for taking the time to review our manuscript and for your comments. They are crucial and valuable to us in raising the quality standard of our work.

We wanted to inform You that we have made a general revision of the English and grammar. In addition, a specification for Your revisions is below:

-"It should be stated that the differentiation between HGSS and undifferentiated uterine sarcoma is only genetic. A histological diagnosis in such cases is not enough. In the past, the majority of the diagnosed undifferentiated uterine sarcomas were actually HGSS. Therefore, the articles which investigated undifferentiated uterine sarcomas and you cited ( references -7, 15) are old and this could not be undifferentiated uterine sarcomas, but HGSS."

Thank you for this Your observation, and we agree with you that this should be emphasized in the work. That is why we specified in the discussion this possible bias (line: 334-337)

-" Use the final sentence of the conclusion at the end of the abstract - "Nonetheless, there are not enough cases in the literature to recommend this procedure"."

We followed your advice and changed it according to your observation

- "Please change the references list according to the MDPI guidelines"

We followed your advice and changed it according to your observation

Also, you can find the rewritten and corrected version of the manuscript in the attached file. We highlighted any changes made.

Thank you very much for your advice and comments. We hope we have complied with your requests.
